# cigFacies: a massive-scale benchmark dataset of seismic facies and its application

Hui Gao[1], Xinming Wu[1], Xiaoming Sun[2], Mingcai Hou[3,4], Hang Gao[1], Guangyu Wang[1], and Hanlin Sheng[1]

[1]School of Earth and Space Sciences, University of Science and Technology of China, Hefei, China;
[2]Institute of Advanced Technology, University of Science and Technology of China, Hefei, China;
[3]Institute of Sedimentary Geology, Chengdu University of Technology, Chengdu, China;
[4]State Key Laboratory of Oil and Gas Reservoir Geology and Exploitation, Chengdu University of Technology, Chengdu, China;

**Correspondence:** Xinming Wu (xinmwu@ustc.edu.cn)

**Abstract.** Seismic facies classification is crucial for seismic stratigraphic interpretation and hydrocarbon reservoir characterization but remains a tedious and time-consuming task that requires significant manual effort. The data-driven deep learning approaches are highly promising to automate the seismic facies classification with high efficiency and accuracy, as they have already achieved significant success in similar image classification tasks within the field of computer vision (CV). However, unlike the CV domain, the field of seismic exploration lacks a comprehensive benchmark dataset for seismic facies, severely limiting the development, application, and evaluation of deep learning approaches in seismic facies classification. To address this gap, we propose a comprehensive workflow to construct a massive-scale benchmark dataset of seismic facies and evaluate its effectiveness in training a deep learning model. Specifically, we first develop a knowledge graph of seismic facies based on the geological concepts and seismic reflection configurations. Guided by the graph, we then implement three strategies of field seismic data curation, knowledge-guided synthesization, and GAN-based generation to construct a benchmark dataset of 8000 diverse samples for five common seismic facies. Finally, we use the benchmark dataset to train a network and then apply it on two 3-D seismic data for automatic seismic facies classification. The predictions are highly consistent with expert interpretation results, demonstrating the diversity and representativeness of our benchmark dataset is sufficient to train a network that can generalize well in seismic facies classification across field data. We have made this dataset, the trained model and associated codes publicly available for further research and validation of intelligent seismic facies classification.

## 1 Introduction

Seismic facies classification aims to delineate individual units based on the specific reflection characteristics (e.g. reflection configuration, continuity, amplitude and frequency contents), which is a fundamental and essential step in the seismic stratigraphic analysis and contributes to the interpretation of sedimentary environments and hydrocarbon reservoir distributions (Sheriff, 1976; Sangree and Widmier, 1977; Veeken, 2006; Jia and Zhao, 2007; Xu and Haq, 2022). With the dramatic increase in the amount of 3-D seismic data, manual interpretation method is typically labor-intensive and heavily relies on the

experienced experts, thus the automatic seismic facies classification is the trend. Moreover, the development of automatic seismic facies classification approaches benefits the accurate and efficient analysis of depositional environments and lithologic distributions.

In recent years, many methods have been proposed for automatic seismic facies classification by using supervised, semi-supervised and unsupervised learning. Supervised learning methods (Wrona et al., 2018; Zhao, 2018; Liu et al., 2018; Zhang et al., 2021) first use large amounts of labeled data to train a CNN model, and then use the trained model for automatic seismic facies classification. Semi-supervised learning methods (Qi et al., 2016; Dunham et al., 2020; Liu et al., 2020) use both labeled and unlabeled data to train the network to learn the features and distributions characterizing seismic facies. Unsupervised

learning methods (Qian et al., 2018; Zhao et al., 2018; Duan et al., 2019; Puzyrev and Elders, 2022; Li et al., 2023) first extract the nonlinear, discriminant and invariant features from unlabeled data, and then cluster or classify these features for automatic seismic facies classification. The supervised learning methods often exhibit weak generalization capabilities across different surveys due to a lack of labeled samples, while semi-supervised and unsupervised methods frequently encounter issues with high uncertainty in prediction results. Besides, seismic facies can be classified into several different categories based on

different attribute parameters, which leads to challenges in the construction of seismic facies datasets and the assessment of the results.

To solve these problems, developing a knowledge graph of seismic facies and using it to provide guidelines for constructing a benchmark dataset is considered an effective methodology. Knowledge graph is a graphical representation model consisting of entities (nodes) and relationships (edges), which aims to represent knowledge in the form of graphs (Paulheim, 2017; Fensel

et al., 2020; Hogan et al., 2021). Currently, knowledge-driven geoscience big data researches have been successfully applied in various kinds of geoscience data-mining tasks (Zhou et al., 2021; Ma et al., 2023; Zhang et al., 2023; Hu et al., 2023). In this work, we construct a knowledge graph of seismic facies, grounded in geological concepts and seismic reflection patterns. This graph guides our processes of data selection, label generation, analysis, and result assessment.

Currently, the construction of the dataset primarily relies on manually interpreted field data and labelled synthetic data. To

address the lack of representative benchmark datasets for seismic facies and improve its automatic classification, we implement a comprehensive workflow of three strategies (field data curation, knowledge-guided synthesization and Generative Adversarial Network (GAN)-based generation) shown in left blue box in Fig. 1 to construct a massive-scale, feature-rich and high-realism benchmark dataset of seismic facies and use it to train a CNN model for accurate and efficient seismic facies classification shown in right red box in Fig. 1.

## 2    Building a massive-scale benchmark dataset of seismic facies

In this section, we initially construct a knowledge graph of seismic facies based on the geological concepts and seismic facies configurations. Guided by the graph, we develop three strategies to construct the benchmark dataset of seismic facies (Fig. 1). The first strategy is to build field samples from field data curation with raw data collection, data standardization and skele-

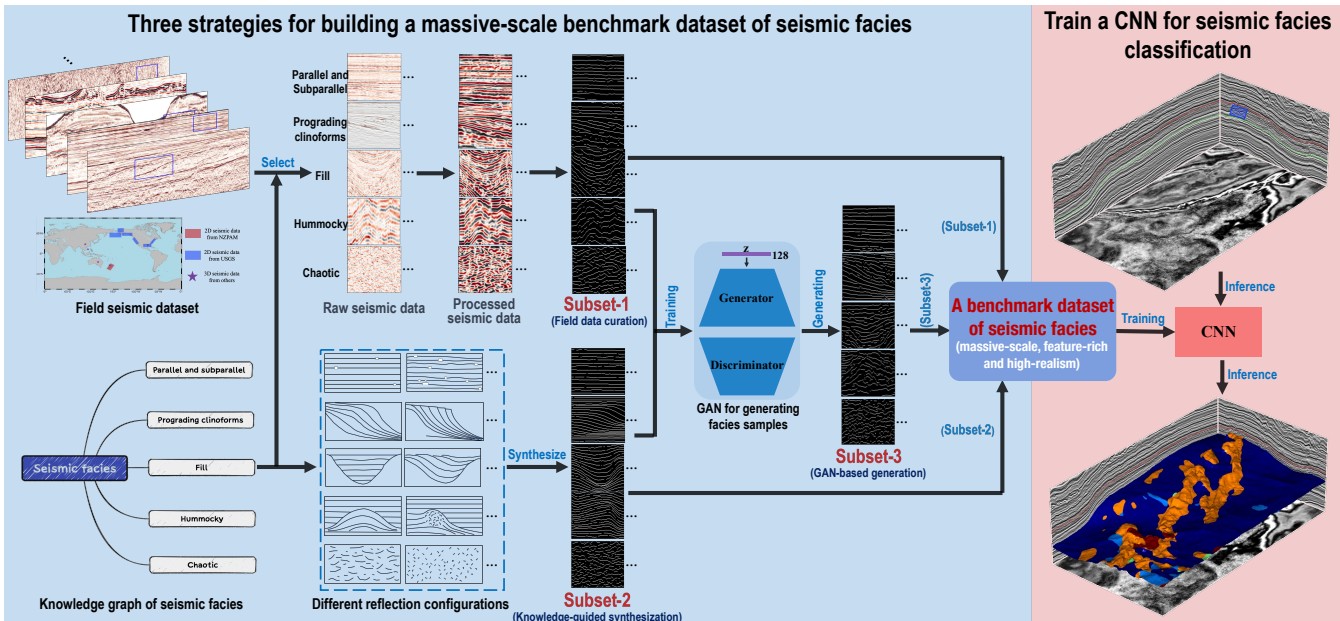

**Figure 1.** The workflow of constructing a massive-scale, feature-rich and high-realism benchmark dataset of seismic facies (blue box) and deep learning for seismic facies classification (red box). We first develop a knowledge graph of seismic facies based on geological concepts and seismic facies configurations. Guided by the graph, we implement three strategies of field seismic data curation, knowledge-guided synthesization, and AI-based generation to construct a massive-scale benchmark dataset. Finally, we use the benchmark dataset to train a CNN model and then apply it on 3-D field seismic data for automatic seismic facies classification.

tonization processes. The second strategy is to build synthetic samples from knowledge-guided synthesization by synthesizing
geological structural curves. The final strategy is to build synthetic samples from AI-based generation with GAN model.

## 2.1 Knowledge graph of seismic facies

Before constructing the massive-scale benchmark dataset of seismic facies, it is necessary to develop a knowledge graph of seismic facies based on the geological concepts and seismic reflection configurations, which can provide guidelines for preparing representative dataset samples and assessing facies classification results. Based on specific seismic reflection configurations,
seismic facies can be roughly divided into parallel and subparallel, prograding clinoforms, fill, hummocky, chaotic, divergent, wave and reflection free (Mitchum Jr et al., 1977a, b; Veeken, 2006; Xu and Haq, 2022) (Fig. 2). Besides, these seismic facies can be further subdivided based on several independent parameters such as the reflection configurations, continuity, amplitude and frequency. For example, parallel and subparallel reflection can be subdivided into 27 different types based on the frequency (high, middle and low), amplitude (strong, moderate and week) and continuity (excellent, medium and poor). Based
on different reflection patterns, prograding clinoforms, fill and hummocky can be futher subdivided into five (sigmoid, oblique,

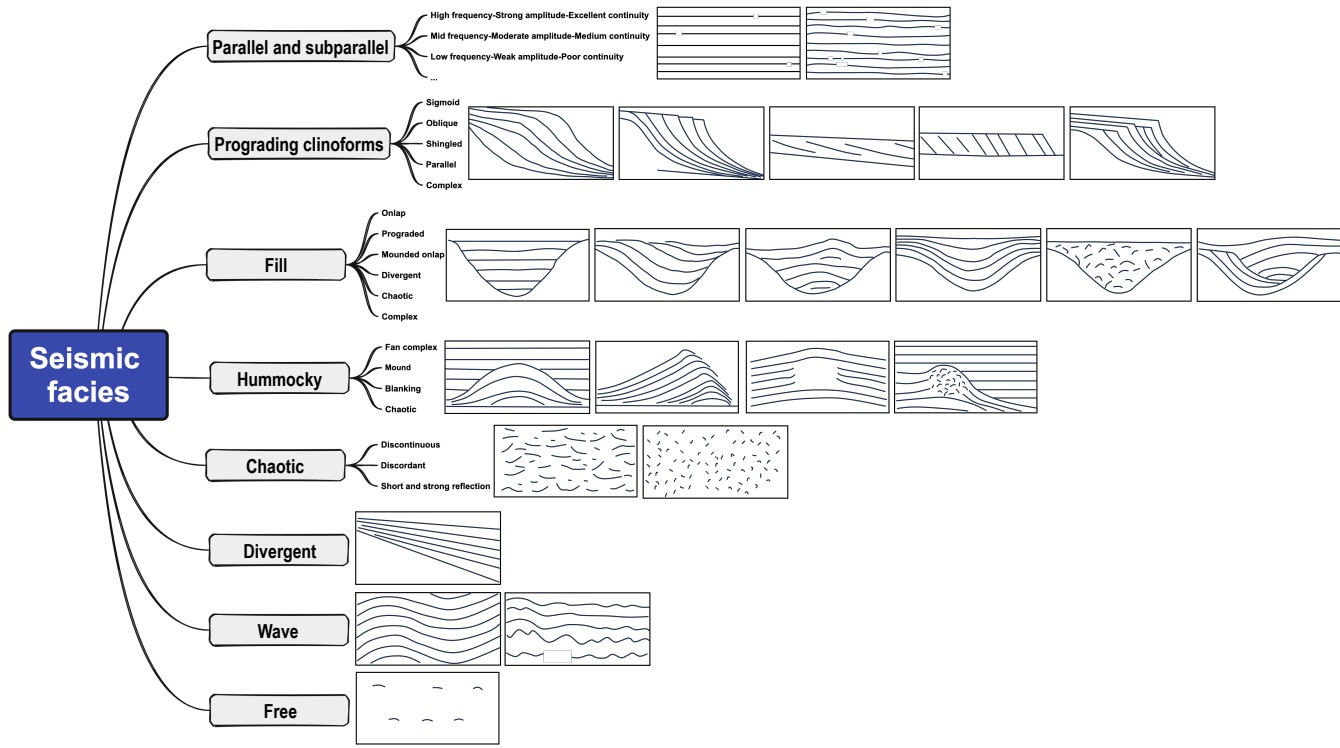

**Figure 2.** Knowledge graph of seismic facies and corresponding typical seismic reflection configurations (modified from Mitchum Jr et al. (1977a, b); Xu and Haq (2022)). In this graph, we roughly divided the seismic facies into eight types (parallel and subparallel, prograding clinoforms, fill, hummocky, chaotic, divergent, wave and reflection free) based on specific seismic reflection configurations. Besides, we also subdivided these seismic facies based on several independent parameters such as the reflection configurations, continuity, amplitude and frequency, and illustrate the typical seismic reflection configurations for each type of seismic facies.

shingled, parallel and complex), six (onlap, prograded, mounded onlap, divergent, chaotic and complex) and four (fan complex, mound, blanking and chaotic) types, respectively.

As shown in Fig. 2, we develop a knowledge graph of seismic facies and illustrate the typical seismic reflection configurations for eight types of seismic facies. However, considering the requirement for data amount and diversity in this work, we take the five most common seismic facies (parallel and subparallel, prograding clinoforms, fill, hummocky, and chaotic) as an example to explain how to construct a massive-scale, feature-rich and high-realism benchmark dataset of seismic facies from field data curation, knowledge-guided synthesization and GAN-based generation.

### 2.2 Building facies samples by field data curation

We start building our benchmark dataset by employing the field seismic data curation strategy with a series of steps including raw data collection, manual interpretation and classification, bandpass filtering, resampling, amplitude equalization, and

skeletonization. We first collect almost 4000 global publicly available 2-D seismic profiles and 10 3-D seismic data from the sources of United States Geological Survey (USGS), New Zealand Petroleum And Minerals (NZPAM), South Australian Resources Information Gateway (SARIG), Society of Exploration Geophysicists (SEG) and so on. These 2-D and 3-D seismic data amount to around 130G, primarily located in the Gulf of Mexico, East and West Coast of America, Alaska, Bering Sea, Beaufort Sea, New Zealand, Southern Australia and Sichuan Basin (see the data distributions map in Figure 1).

We then manually select, crop and classify these field seismic data based on the knowledge graph (Fig. 2). As shown in the raw seismic data of Fig. 3, we totally collect 1000, 700, 500, 500 and 700 2-D raw seismic data for five common seismic facies, respectively. However, due to the different data sources, depositional environments and data processing methods, these raw seismic data have large differences in sampling rates, amplitude and frequency distributions (as shown in Fig. 3 and Fig. 4a) among same and different classes of seismic facies. These data variations and uncertainties are not related to the seismic facies. Moreover, they may pose significant interference to deep learning models in learning the crucial features such as texture patterns and reflection configurations, which are essential for identifying seismic facies categories. To migrate such uncertainties in building our standard benchmark dataset, we introduce the data standardization process (Fig. 4) for each raw seismic data, including filtering, resampling, amplitude equalization, frequency equalization and so on. After applying the data standardization process, the processed seismic data have been significantly improved in the consistency of the sampling rates, amplitude and frequency distributions (as shown in Fig. 3b and Fig. 5). Finally, we retain the main geological structure informations of strata by keeping only the waveform peaks as ones and setting elsewhere zeros to obtain the corresponding field skeletonization images shown in Fig. 3c and Fig. 6.

Compared to the skeletonization images (Fig. 4d) obtained directly from raw seismic data, the ones (Fig. 4c) with data standardization can more clearly reflect the geological structure characteristics and enhance the consistency among the same and different classes of seismic facies. The whole curation strategy, perticularly the data standardization processes and skeletonization, eliminates uncertainties inherent in field data from various surveys. This approach retains only the texture patterns associated with seismic facies to produce standardized images for constructing the benchmark dataset. The same processing techniques will also be applied to inference data to ensure that a deep learning model trained on this dataset achieves consistent predictions.

However, the facies samples from only these field seismic datasets are imbalanced in categories and lack diversity and therefore are not sufficient to build a massive-scale and representative benchmark dataset. For example, parallel and subparallel data are more common than fill or hummocky data in field seismic data. Additionally, some specific patterns (e.g., parallel prograding clinoforms, chaotic fill, complex fill and blanking hummocky) are rare in these publicly available field seismic data.

## 2.3 Building facies samples from knowledge-guided synthesization

In order to overcome the sample imbalance and improve the diversity of the dataset, we further develop the second strategy to automatically generate synthetic facies samples based on the knowledge graph of seismic facies and independent seismic

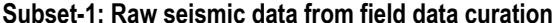

**Subset-1: Raw seismic data from field data curation**

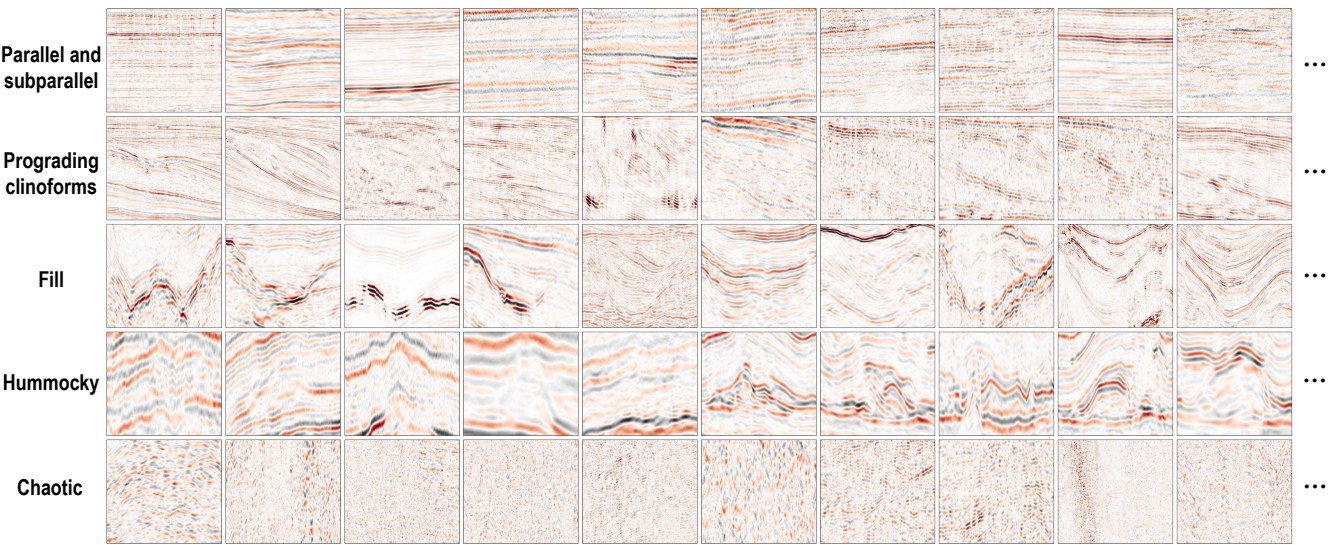

**Figure 3.** Subset-1: raw seismic data manually collected and interpreted from the large amount of publicly available seismic datasets. In total, we select, crop and classify 1000, 700, 500, 500 and 700 2-D raw seismic data for parallel and subparallel, prograding clinoforms, fill, hummocky and chaotic, respectively.

reflection configurations. We first define different geological structural curves by using following geometric functions:

$$z = z_0, \tag{1}$$

$$z = k_0 \cdot x + z_0, \tag{2}$$

$$z = k_0 \cdot x^2 + z_0, \tag{3}$$

$$z = \frac{1}{k_1 + k_2 \cdot e^{-k_3 \cdot x}}, \tag{4}$$

$$z = \frac{e^{k_1 \cdot x} - e^{-k_2 \cdot x}}{e^{k_1 \cdot x} + e^{-k_2 \cdot x}}, \tag{5}$$

where $x$ and $z$ represent the position in the crossline and depth directions, respectively. Other parameters ($z_0$, $k_0$, $k_1$, $k_2$ and $k_3$) are used to control the geometry and distribution of the geological structural curves. We then set different combinations of geological structural curves based on different seismic facies categories. Additionally, we randomly set shape parameters for these geological structural curves and combine them at random intervals to enhance their diversity. Furthermore, we can also first define some key points for some complex geological structures, and then generate the corresponding geological structural curves by applying interpolation process. After generating these different geological structural curves, we add random noise and apply random local mask to each curve for improving the realism of the synthetic curves. Finally, we set ones on the geological structural curves and zeros elsewhere to generate the corresponding synthetic skeletonization data.

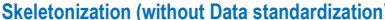

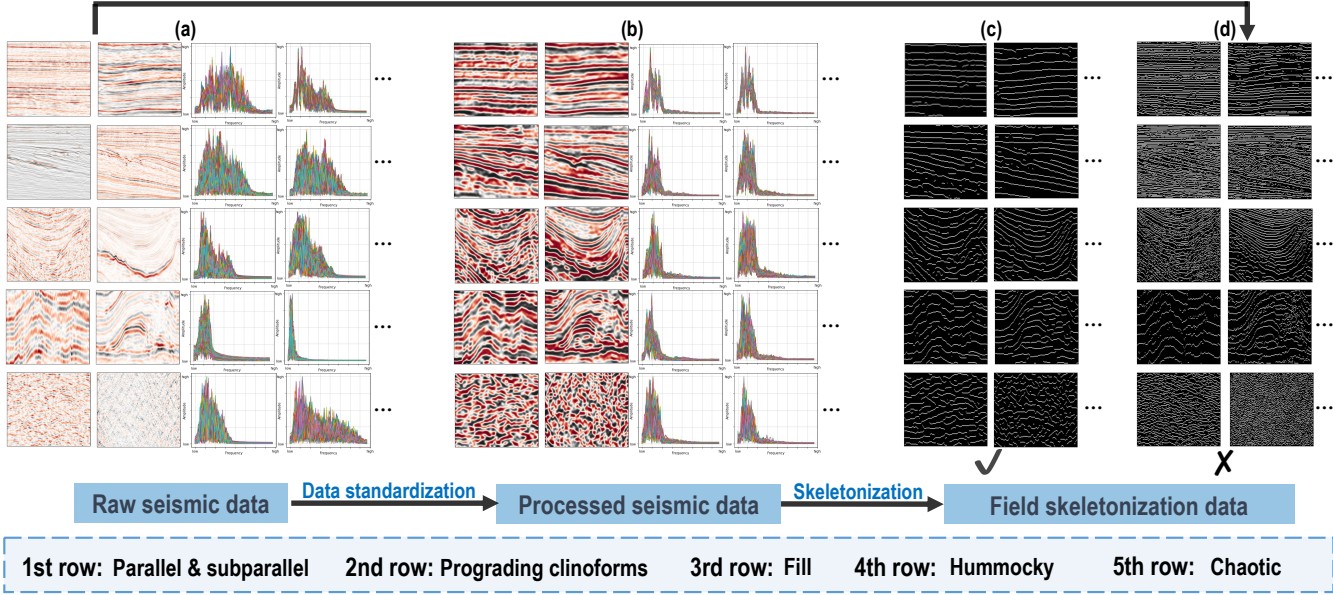

**Figure 4.** The workflow of constructing the field samples from field seismic data curation. We first manually collect and interpret raw seismic data (a). Then we introduce data standardization process for each raw seismic data to improve the consistency of the sampling rates, amplitude and frequency distributions. After obtaining the processed seismic data (b) , we retain the main geological structure informations of strata by keeping on the waveform peaks as ones and setting elsewhere zeros to obtain the corresponding field skeletonization images (c). Furthermore, the skeletonization images (d) obtained directly from raw seismic data (without standardization) exhibit significant uncertainties and variations.

In this way, we randomly generate synthetic facies samples for each types of seismic facies, especially some specific patterns which are rare in field data curation, thus complementing the benchmark dataset of seismic facies. Finally, we automatically generate 500 synthetic facies samples for five common seismic facies shown in Fig. 7, respectively. Compared to the field facies samples shown in Fig. 6, the synthetic facies samples generated from knowledge-guided synthesization contain more diverse patterns and reduce sample imbalance. However, these synthetic facies samples may be ideally patterned and lack realism.

## 2.4 Building facies samples from GAN-based generation

As shown in the subset-1 and subset-2 in Fig. 6, Fig. 7 and Fig. 8a,b, the field facies samples has high realism but poor diversity, while the synthetic facies samples has strong diversity but poor realism. In order to construct the comprehensive benchmark dataset of seismic facies, we develop the final strategy of GAN-based generation (Fig. 8) to build more facies samples with both strong diversity and high realism.

As shown in Fig. 8c, the architecture of deep learning network used in this work is modified from the progressive growing of GANs proposed by Karras et al. (2017). Traditionally, the progressive growing of GANs consists of a generator model (G) and

**Subset-1: Processed seismic data from field data curation**

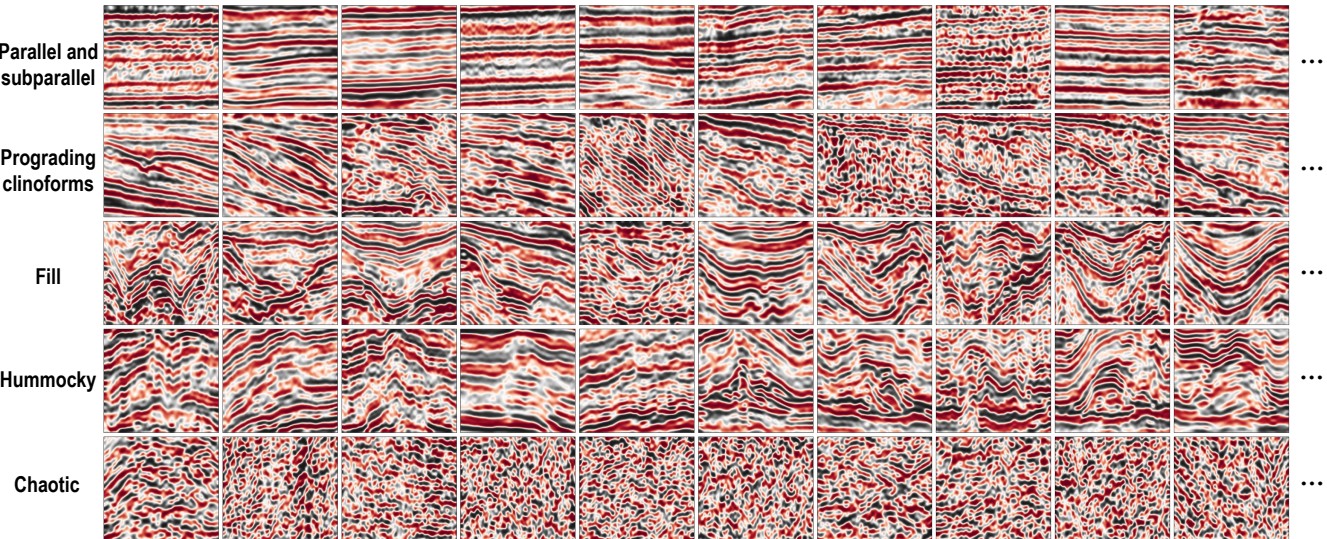

Parallel and subparallel … 
Prograding clinoforms … 
Fill … 
Hummocky … 
Chaotic … 

**Figure 5.** Subset-1: processed seismic data generated from raw seismic data by applying the data standardization processes. Compared to the raw seismic data, the processed seismic data exhibit significant improvement in the consistency of the sampling rates, amplitude and frequency distributions.

a discriminator model (D), where G was used to capture the data distribution and generate fake images to resemble the training dataset (real images), and D was used to assess the probability that images are real or fake. The G is composed of a Gen-1 module, five Gen-2 modules and a $Conv_{1\times1}$ layer, where Gen-1 module consists of a $4 \times 4$ convolutional layer and a $3 \times 3$ convolutional layer, and Gen-2 module consists of an upsampling layer and two $3 \times 3$ convolutional layers. The D is composed of a $Conv_{1\times1}$ layer, five Dis-1 modules and a Dis-2 module, where Dis-1 module consists of two $3 \times 3$ convolutional layer and a average pooling layer, and Dis-2 consists of a minibatch stddev layer, a $3 \times 3$ convolutional layer, a $4 \times 4$ convolutional layer, a flatten layer and a linear layer. Compared to traditional GANs, the progressive growing of GANs does not directly generate high-resolution images, but starts from generating simple low-resolution images and then continuously increases the resolution of the generated images during the network training. This training strategy allows the network to learn the features of the training dataset from coarse to fine scales, resulting in faster training speed, higher stability and better quality images. Besides, we use WGAN-GP loss proposed by Gulrajani et al. (2017) as the GANs loss function $\mathcal{L}(G, D)$ to optimize the network.

We combine subset-1 and subset-2 as training datasets to train the progressive growing of GANs. Initially, we first train a simple network consisting of a Gen-1 module, two $Conv_{1\times1}$ layer and a Dis-2 module to generate and access the real and fake facies samples with $4 \times 4$ scale. After stabilizing the training of this simple network, we then incorporate a Gen-2 module and a Dis-1 module into it for doubling the resolution of G and D. In this way, our network will progressively grow to steadily generate high resolution ($128 \times 128$) facies samples. Finally, we use the trained G to automatically generate 500

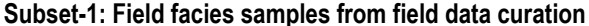

**Subset-1: Field facies samples from field data curation**

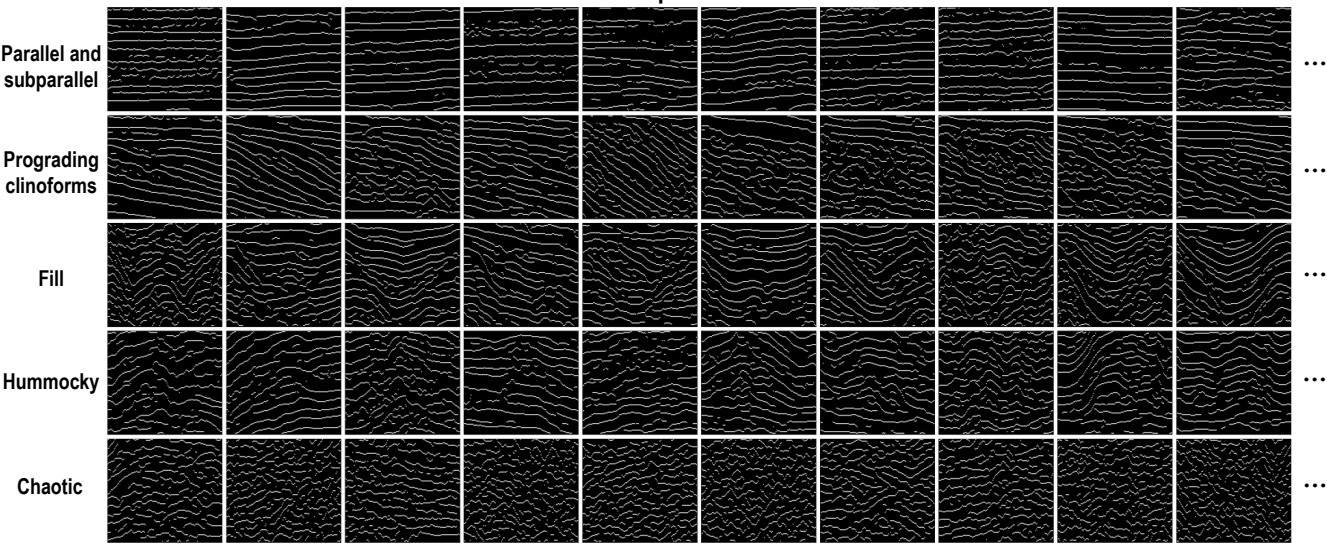

**Figure 6.** Subset-1: field facies samples generated from processed seismic data by applying the skeletonization process. After obtaining the processed seismic data, we retain the main geological structure informations of strata to generate the corresponding field skeletonization images. Finally, we use the first strategy to manually select 1000, 700, 500, 500 and 700 field facies samples for five common seismic facies, respectively.

facies samples for each type of seismic facies shown in subset-3 in Fig. 8d and Fig. 9. Compared to the subset-1 and subset-2, the facies samples constructed by the GAN-based generation hold both high diversity and strong realism.

## 2.5 The final benchmark dataset of seismic facies

After applying three strategies of field data curation, knowledge-guided synthesization and GAN-based generation to build
diverse facies samples, we construct a massive-scale, feature-rich and high-realism benchmark dataset of seismic facies and we display some facies samples in Fig. 6-9. As shown in Fig. 10, we finally generate a total of 2000, 1500, 1500, 1500, and 1500 diverse facies samples (128[inline]×128[time]) for five common seismic facies (parallel and subparallel, prograding clinoforms, fill, hummocky and chaotic), respectively. The final benchmark dataset, named cigFacies, has been made publicly available at https://zenodo.org/records/10777460 (Gao et al., 2024a).

**3 Deep learning for seismic facies classification**

After constructing the comprehensive benchmark dataset of seismic facies (Fig. 10), we use it to train a simple CNN for the seismic facies classification task shown in right red box in Fig. 1. In this study, we first use 6400 samples to train the model

**Subset-2: Synthetic facies samples from knowledge-guided synthesization**

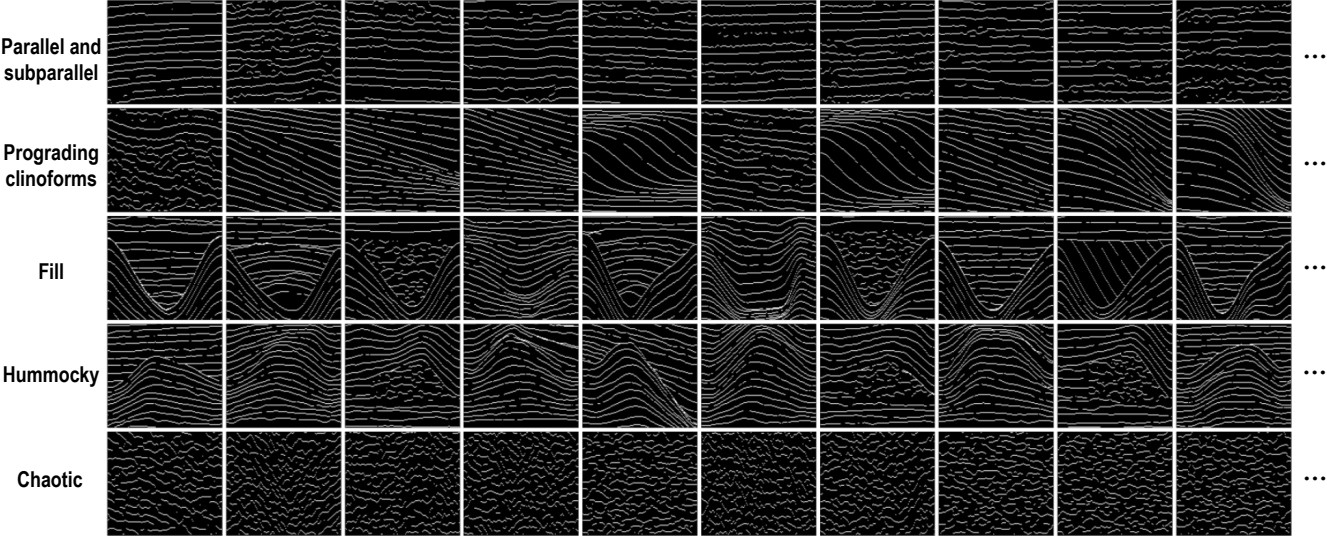

**Figure 7.** Subset-2: synthetic facies samples generated from knowledge-guided synthesization. In this strategy, we first construct some geological structural curves from geometric functions or interpolation process. Then we add random noise and mask for each curve to improve the realism of synthetic facies samples. Finally, we use the second strategy to automatically generate 500 synthetic facies samples with more diverse patterns for each seismic facies, respectively.

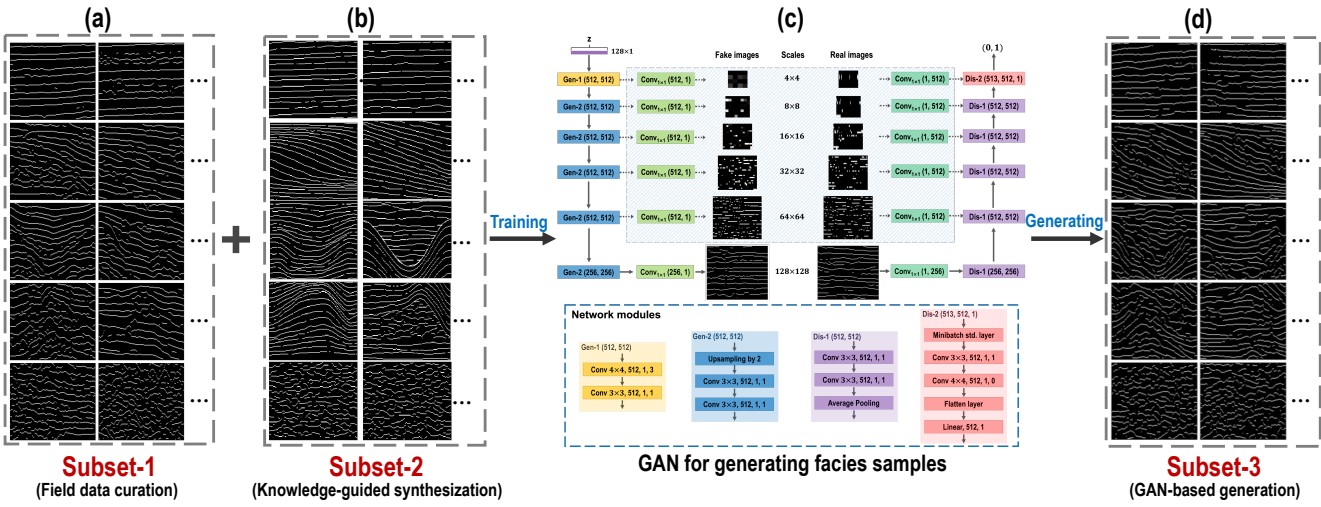

**Figure 8.** The workflow of constructing the synthetic samples from GAN-based generation. In this strategy, we first use the subset-1 (a) and subset-2 (b) generated from the first and second strategies to train a progressive growing of GANs (c), and then use the trained G to automatically generate synthetic facies samples (d) for each type of seismic facies.

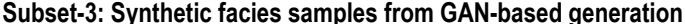

**Subset-3: Synthetic facies samples from GAN-based generation**

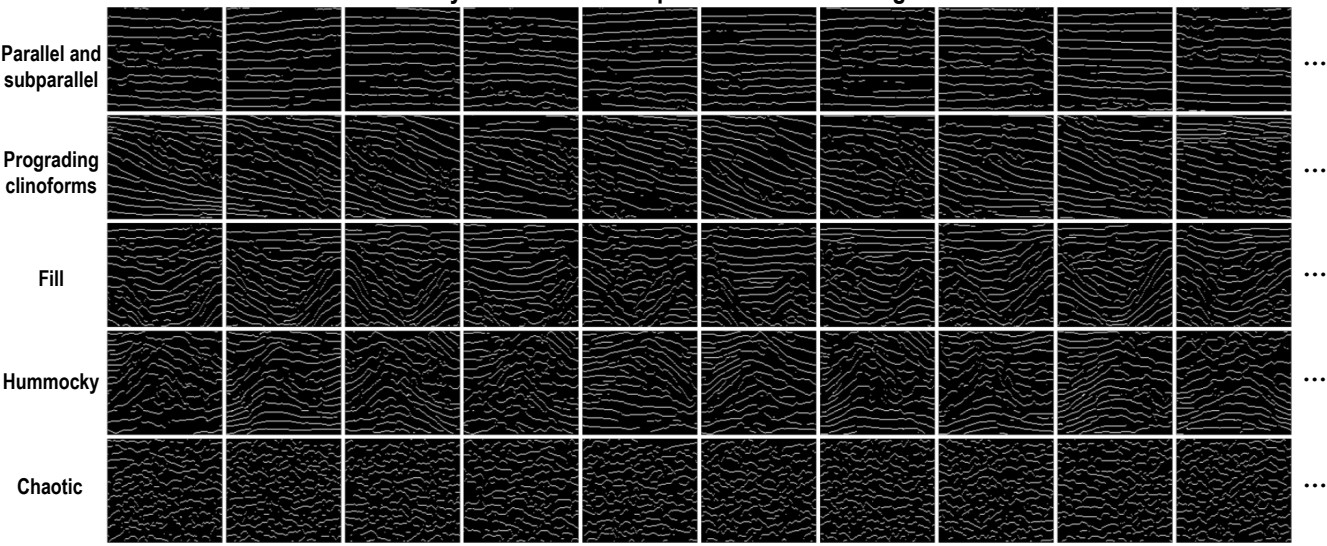

**Figure 9.** Subset-3: synthetic facies samples generated from GAN-based generation. In total, we use the third strategy to automatically generate 500 synthetic facies samples with both high diversity and strong realism for each type of seismic facies, respectively.

and another 1600 samples for validation. Then we develop a predicted workflow to apply the trained network for automatic seismic facies classification in the 3-D field seismic data.

## 3.1 Training and Validation

We consider seismic facies classification as an image classification problem with the goal to classify the 3-D field seismic data to the corresponding seismic facies (e.g. parallel and subparallel, prograding clinoforms, fill, hummocky and chaotic). In this study, we use a simple deep learning network (ResNet-50) proposed by He et al. (2016) (Fig. 11a) to implement automatic seismic facies classification. We train and validate our CNN model by using 6400 and 1600 random pairs of facies samples. Besides, in order to improve the diversity of dataset, we apply random data augmentation strategies (e.g. flip, translation, crop and resize) for each facies sample before feeding it into the network. we train our network by using the following cross entropy loss function $\mathcal{L}$:

$$\mathcal{L} = -\sum_{i=0}^{N-1} y_i \log(x_i), \tag{6}$$

where $N$ denotes the number of classes, and $x_i$ and $y_i$ represent the one-hot prediction and label at the $i$-th class, respectively. Considering the computation time and memory, we set the batch size to 32 and use the Adam optimizer to optimize the network parameters. In the training process, we start the learning rate at 0.01 and adaptively reduce the learning rate by half when the training metric stagnates within 2 epochs. As shown in Fig. 11 b, c, both the training loss and validation loss converge to 0.006 and 0.1, while the learning rate decreases from 0.01 to 0.00001 after 200 epochs.

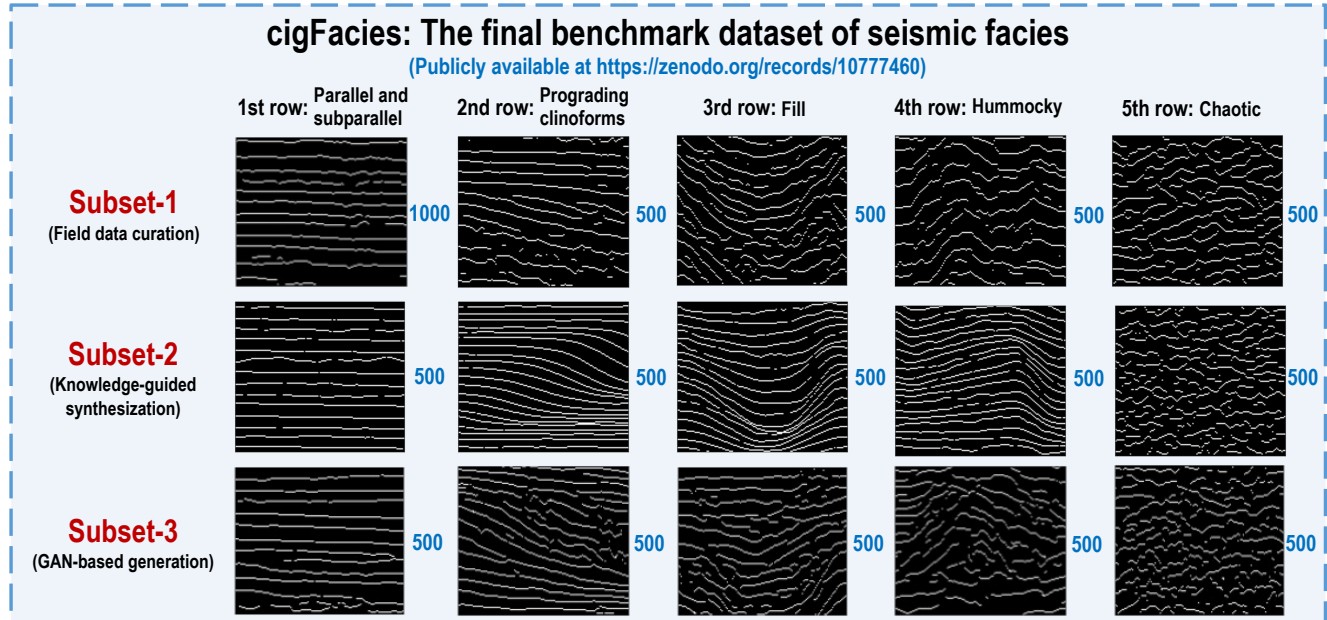

**Figure 10.** cigFacies: the final benchmark dataset of seismic facies construct from three strategies of field data curation, knowledge-guided synthesization and GAN-based generation. In this dataset, we totally generate 2000, 1500, 1500, 1500, and 1500 diverse facies samples (128[inline]×128[time]) for five common seismic facies (parallel and subparallel, prograding clinoforms, fill, hummocky and chaotic), respectively.

To verify the performance of the trained network, we first apply it to the validation dataset which are not included in training dataset. As shown in the Fig. 11 d, the predicted results are highly consistent with the labels. Besides, the predicted accuracy for five common seismic facies in validation dataset can up to 97.75%, 99%, 99.67%, 97.33% and 98.33%, which indicates that the trained network has successfully learned for automatic seismic facies classification.

## 3.2 Testing on the 3-D field seismic data

To further verify the performance of the trained model, we develop a predicted workflow for automatic seismic facies classification in 3-D field seismic data shown in Fig. 12a. We first use an automatic horizon-picking method (Wu and Fomel, 2018) to extract the top and bottom surfaces (green and red curves in Fig. 12b) of the target section in 3-D field seismic data. Then we flatten the field seismic data with the bottom surface to eliminate the influence of the geological structures. Besides, we set a sliding window (blue box in Fig. 12b) centered on the midpoint of the top and bottom horizons and bounded by these surfaces to extract 2-D raw seismic image pixel by pixel (or trace by trace). The width of the sliding window mainly depends on the size of the classified object within the target section and is typically slightly larger than the width of the objects. We further apply the standardization and skeletonization processes to the flattened image to make it consistent with the training dataset. Finally, we feed the corresponding skeletonization image into the trained network for automatic seismic facies classification.

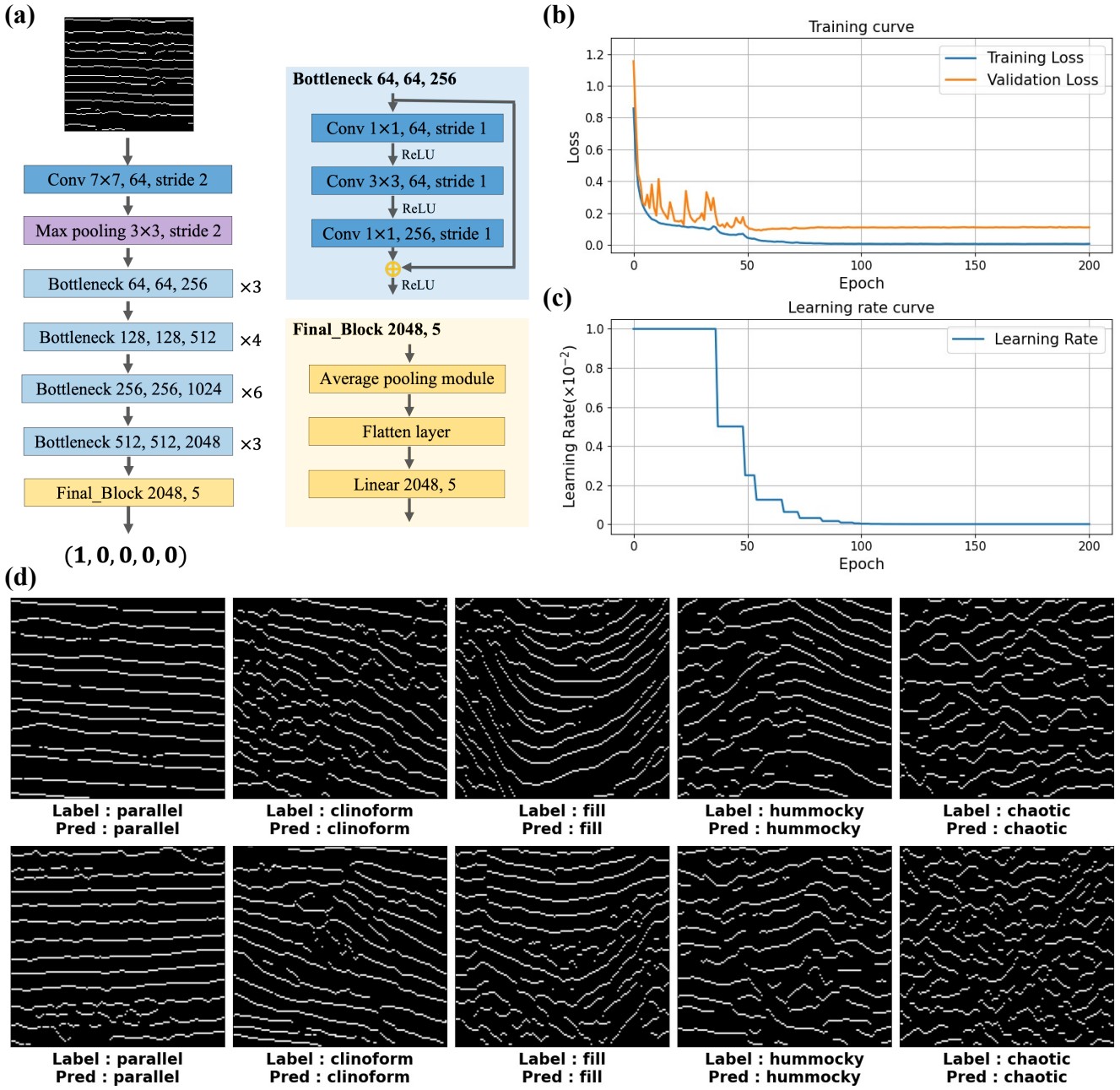

**Figure 11.** (a) The architecture of deep neural network (ResNet-50) used in this work for automatic seismic facies classification. The training (blue) and validation (orange) loss curves (b) and learning rate curve (c) during network training. After training the network, we apply the trained network to the validation dataset to verify its performance. The predicted results are consistent with the labels (d), which demonstrated that the trained network has successfully learned to automatically classify the seismic facies.

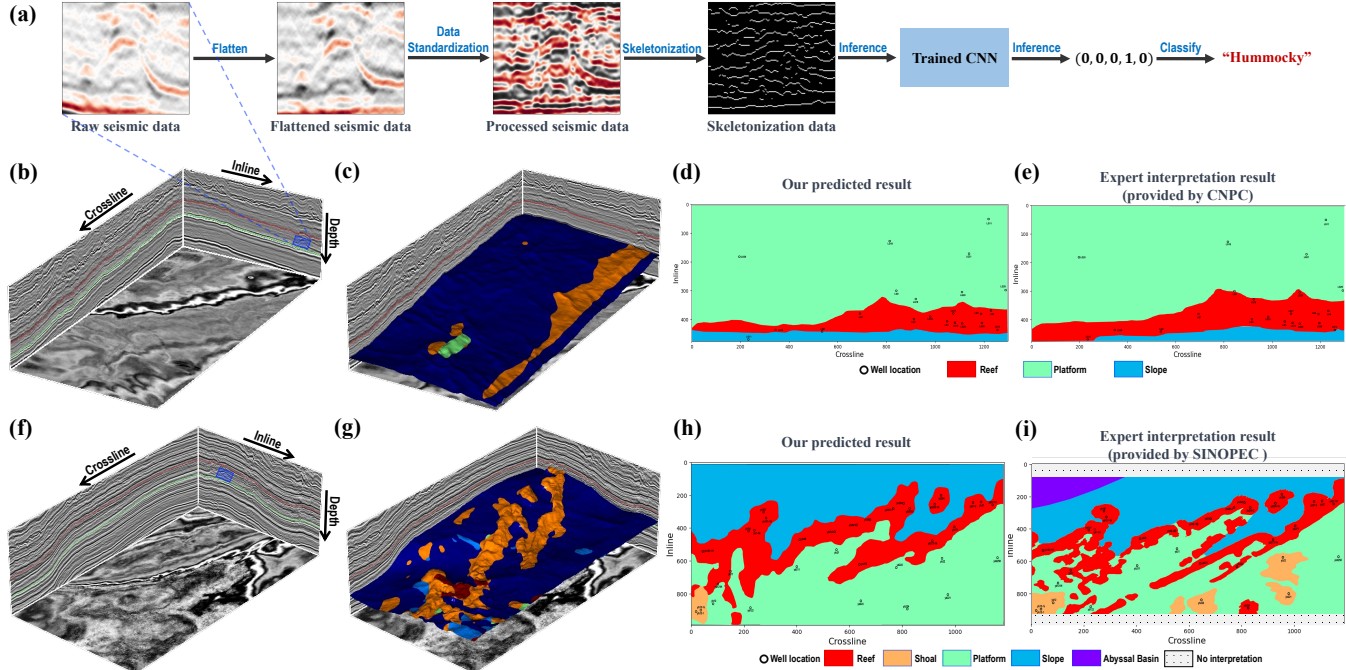

**Figure 12.** We employ the prediction workflow (a) with a sliding window scanning the entire 3D target section in the 3-D seismic data (b and f), yielding the seismic facies classification results (c and g). Then we obtain the corresponding sedimentary facies results (d and h) based on the predicted seismic facies result, well log informations, seismic data and geological and geophysics knowledge. Compared to the expert interpretation results (e and i), our predicted sedimentary facies results are high consistent.

In this work, we apply the trained network on two distinct 3-D field seismic data (Longang and Yuanba) with complex geological structures. The Longgang (LG), Yuanba (YB) areas in Sichuan Basin develop a huge amount of platform margin reef complexes, which have emerged as an important field for oil and gas exploration (Chen et al., 2012; Xu et al., 2015; Tan et al.,
2020). The first study case is the Permian Changhsing Formation of the LG 3-D seismic data (991[inline] × 1187[crossline] × 501[time]) shown in Fig. 12b and Fig. 13a. We employ the predicted workflow (Fig. 12a) with a sliding window traversing the entire 3D target strata, yielding the seismic facies classification result shown in Fig. 12c. Besides, we display the predicted results with different 2-D profiles along the crossline and inline directions in Fig. 13b-i. The regions indicated by the blue arrows
are correctly predicted to the hummocky facies, which are roughly consistent with geological structural uplift in corresponding 2-D seismic profiles. However, some artifacts or inaccurate predictions still appear in some areas indicated by red arrow in Fig. 13f and h, which is mainly due to the incomplete flattening of the strata. As shown in Fig. 13i, we can clearly observe a distinct reef-top interface reflection axis indicated by blue arrows. The trend of this reflection axis indicates that the reef gradually backward along the increasing crossline direction, which closely matches the trends observed in both our predicted
results and the expert interpretation results. Finally, we obtain the corresponding sedimentary facies result (Fig. 12d) based on

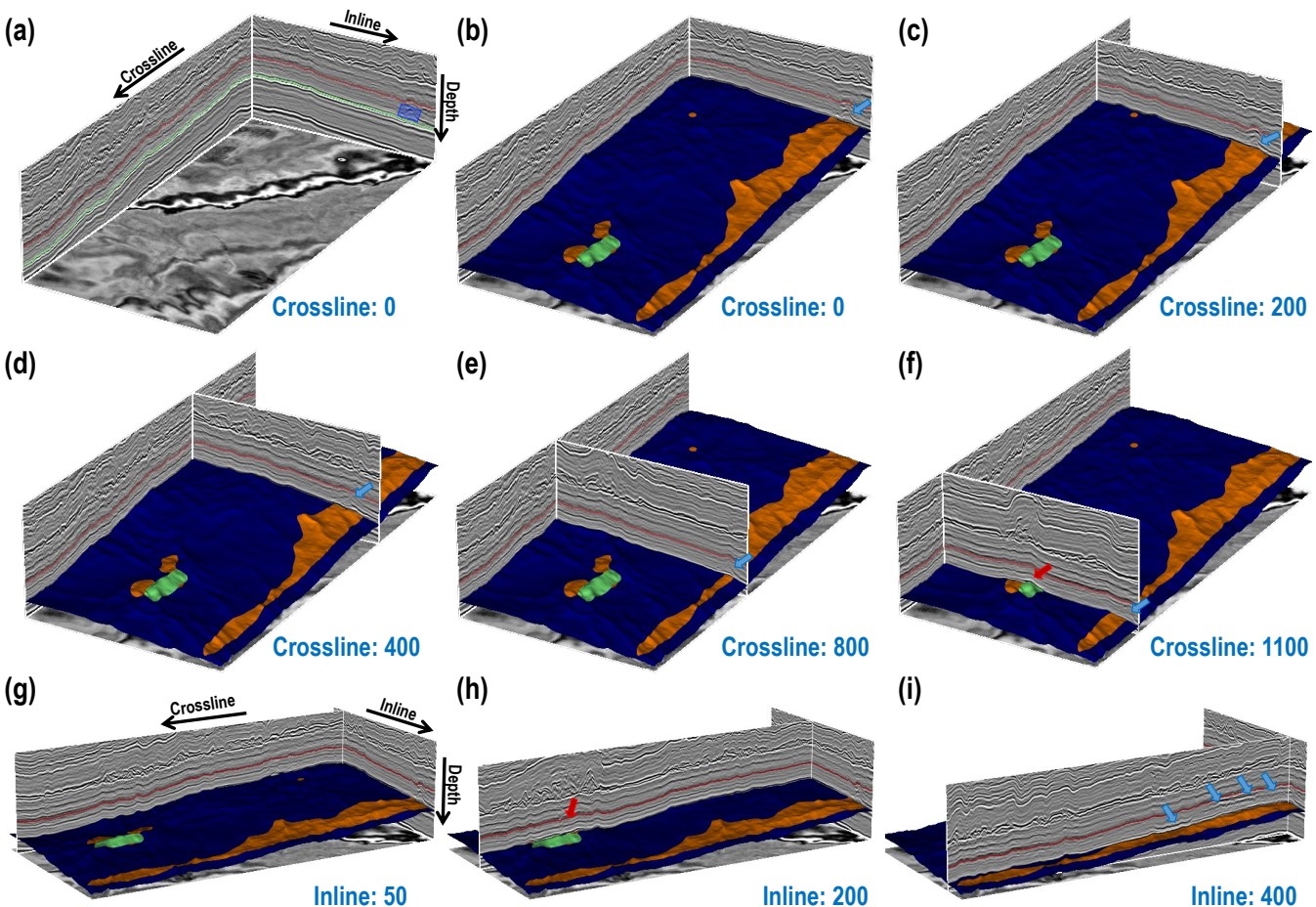

**Figure 13.** (a) 3-D Longgang (LG) seismic data (991[inline]×1187[crossline]×501[time]) and corresponding top and bottom horizons of target strata shown in red and green curves. (b)-(f) Different 2D seismic profile at different crossline direction (0, 200, 400, 800 and 1100) and inline direction (50, 200 and 400), overlaid with the predicted result. The blue arrows indicate areas where predicted results align with the geological structural uplift in different 2D views.

the predicted seismic facies result, well log information, seismic data and geological and geophysical knowledge. Our final sedimentary facies result (Fig. 12d) is highly consistent with the expert interpretation of sedimentary facies shown in Fig. 12e.

The second study case is the Permian Changhsing Formation of the YB 3-D seismic data (1300[inline] × 475[crossline] × 600[time]), as shown in Figs. 12f, 14a. The YB data consists more complex geological structures compared to the LG 3-D
seismic data. Using the same predicted workflow in the previous case, we obtain the corresponding distributions of seismic facies and overlay the result with a manually interpreted horizon shown in Figs. 12g and 14b. The predicted distribution of hummocky seismic facies is consistent with the uplifted areas on the manually interpreted horizon. This high consistency can be also demonstrated in Figs. 14 b-i, which display additional 2D seismic profiles with the predicted result in different 3D view.

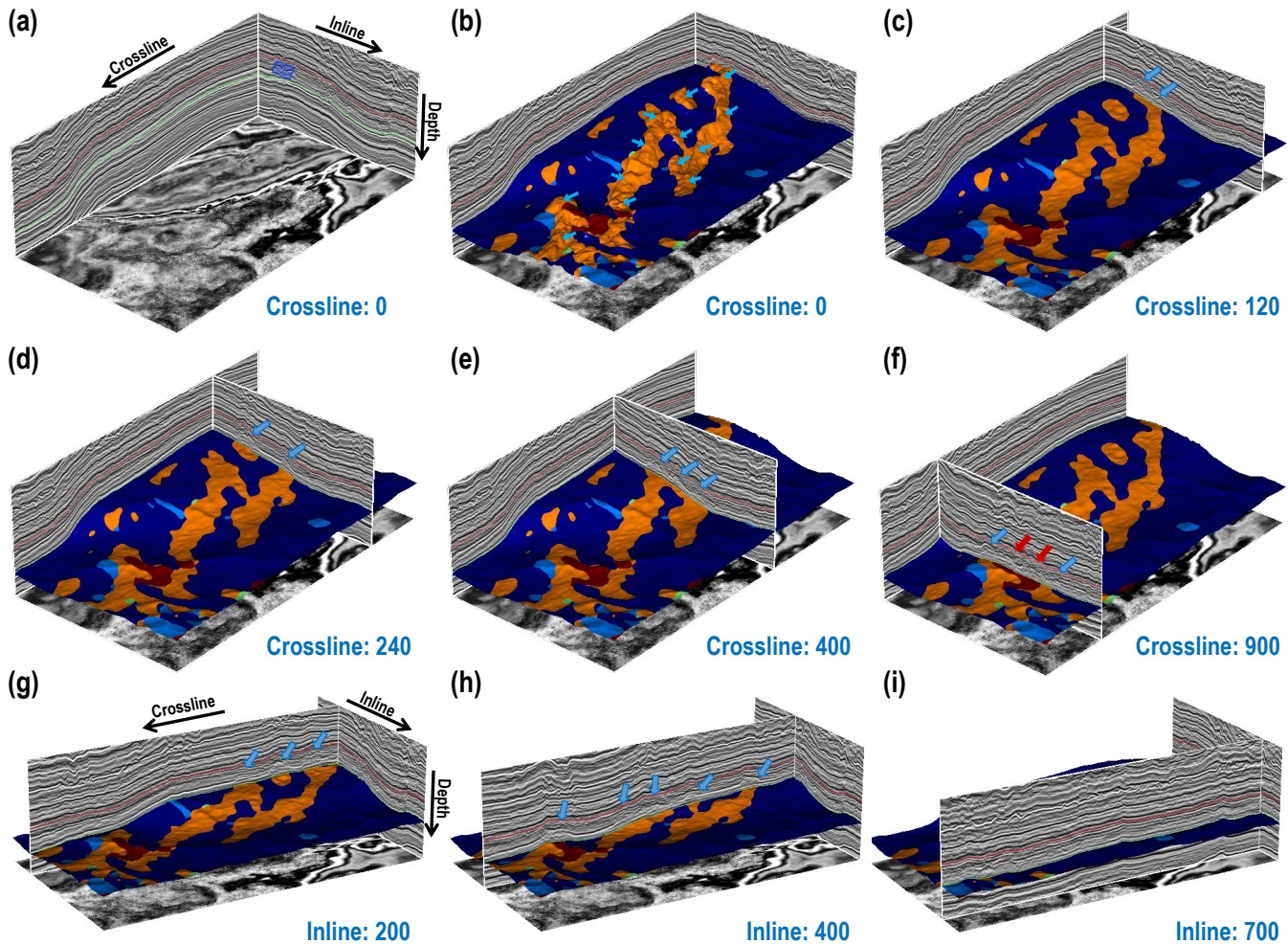

**Figure 14.** (a) 3-D Yuanba (YB) seismic data (1300[inline]×475[crossline]×600[time]) and corresponding top and bottom horizons of target strata shown in red and green curves. (b) 3-D predicted results overlaid on an expert-interpreted horizon, and the distributions of hummocky seismic facie are consistent with the uplifted areas indicated by the blue arrows. (c)-(i) Various 2D views of seismic data with different combinations of crossline (120, 240, 400 and 900) and inline (200, 400 and 700) directions with predicted result.

In particular, the areas indicated by the blue arrows in Figs. 14 g and h demonstrate that our method accurately captures the
distribution of the patch reef and platform reef zone. Moreover, our predicted results are consistent with expert interpretation results of the reef distribution along the complex platform margin. However, some regions indicated by the red arrows (Fig. 14f) are incorrectly classified as other seismic facies, which is probably due to the unsuitable scale of sliding window for these local regions, the influence of boundary effects from sliding window, and incomplete flattening of the stratum. Finally, we also generate the corresponding sedimentary facies results shown in Fig. 12h, where the platform margin reef are clearly and

reasonably resolved and the spatial distribution of the platform margin is highly consistent with the expert interpretation in Fig. 12i.

## 4   Discussion

In this work, we present a benchmark dataset and a deep learning-based approach for automatic seismic facies classification. Our method utilizes a knowledge graph of seismic facies to guide the construction of the dataset, which includes a combination
of field data curation, knowledge-guided synthesization, and GAN-based generation. These strategies avoid the problems such as sample imbalanced, poor diversity and weak realism that usually occur in the traditional dataset construction methods. Besides, the data standardization and skeletonization processes successfully mitigate all potential data uncertainties (not related to seismic facies) across diverse data sources. This enables a deep learning model trained by the dataset to be effectively applied to field data across various surveys, thereby enhancing its generalizability. Applications on the validation dataset and
two distinct 3-D field seismic data (LG and YB) demonstrate that the simple CNN model trained with the benchmark dataset achieves promising performance and strong generalization ability for automatic seismic facies classification task.

Although our method constructs a comprehensive benchmark dataset of seismic facies and achieves promising classification results, some limitations remain in the dataset construction and model application processes. In the data construction processes, we initially develop a knowledge graph primarily categorized by geological structure, emphasizing the role of structural infor-
mation in seismic facies classification. However, the knowledge graph overlooks other important multi-attribute parameters, such as amplitude, continuity, frequency, and wave patterns, which also carry rich information. Additionally, constructing the 2-D skeletonization dataset for seismic facies classification does not fully consider the information contained in the seismic data along the inline direction, which may lead to lateral discontinuities in 3-D applications.

In the model application process, some special geological structure and the introducing of sliding window may also cause the
inaccurate results. The normal or reverse faults in the 3-D seismic data probably introduce unreasonable geological structures when flattening the seismic data, thus resulting in inaccurate predictions. We simplify the classification of 3-D field seismic data by classify the stratigraphic skeletonization information within the 2-D sliding window, without fully utilizing the 3-D information along the inline direction, which may lead to instability in predicted results, especially in the direction perpendicular to the sliding window. Besides, the proper size of sliding window has a significant effect on the results, which needs
to appropriately match the scale of the key seismic facies in field data. Furthermore, due to the predicted result is obtained by scanning pixel by pixel (trace to trace), some inaccurate predicted results may occur in the boundaries between different seismic facies, where the sliding window only contains the partial geological structure.

## 5   Conclusion

We have developed three strategies guided by a knowledge graph to build a benchmark dataset that is vast in scale, rich in
features, and offers high realism. To the best of our knowledge, this dataset is the most extensive dataset of seismic facies cur-

rently available. The seismic facies knowledge graph, developed based on comprehensive literature review, summarizes various typical seismic facies types, along with their corresponding geological origins and seismic response features. This knowledge graph provides comprehensive guidance for the three strategies employed in building the benchmark dataset, ensuring the comprehensiveness and representativeness of the data sample construction. The first strategy of field seismic data curation yields the first subset that is authentic but exhibits some imbalance and limited diversity. The second strategy of sample synthesis, informed by the knowledge graph, generates a second subset of samples containing any category and pattern features, thereby addressing the issues of uneven sample type distribution and lack of diversity in the first subset. However, the synthesized samples also face the problem of being overly idealized and not sufficiently realistic. Consequently, a third strategy, based on AI generation, is adopted to refine the dataset construction. This strategy involves training a GAN model using the already constructed first and second subsets, then leveraging it to derive a third subset with diverse patterns and realistic features. By merging these three subsets, we have ultimately constructed a dataset containing 2000, 1500, 1500, 1500, and 1500 samples for five common seismic facies, respectively. This benchmark dataset has been demonstrated to effectively train a CNN model that achieves notable performance in seismic facies classification across two distinct 3-D field datasets. We have made this benchmark dataset publicly available, encouraging its further enhancement and utilization by others in the development and evaluation of deep learning approaches for seismic facies characterization.

In the future, we can construct a more comprehensive and refined knowledge graph of seismic facies based on multi-attribute parameters such as reflection configurations, continuity, amplitude, frequency, wave pattern and so on. Then we can further construct 3-D seismic datasets with multi-attribute features for more refined seismic facies classification tasks, instead of 2-D skeletonization datasets that only contain structural informations and lack variations along the inline direction. Additionally, we can develop a multi-scale 3-D network for automatic seismic facies classification, which can enhance both the accuracy and stability of predicted results, particularly at different seismic facies boundaries and 3-D field seismic data.

## 6 Code and data availability

The benchmark dataset of seismic facies has been uploaded to Zenodo and are freely available at https://zenodo.org/records/10777460 (Gao et al., 2024a). The corresponding codes for constructing dataset and model training have been uploaded to Zenodo and are freely available at https://zenodo.org/records/13150879 (Gao et al., 2024b).

*Author contributions.* HG, XW, XS and MH initiated the idea of building the benchmark dataset of seismic facies and its application. HG, XW and XS initiated the idea of three strategies to constructing the benchmark dataset of seismic facies. HG, XW, XS, HS and HG conducted the first strategies of field data curation to build field facies samples. HG and XW tested and modified the code for the second and third strategies to build synthetic facies samples. HG carried out the experiments for the training and validation dataset. HG, HG and GW applying the trained network on the field seismic data. XW, XS and MH advised on the benchmark dataset preparation and predicted results analysis from a geological perspective. HG and XW prepared the paper, with contributions from all co-authors.

*Competing interests.* The authors declare that they have no conflict of interest.

*Acknowledgements.* We thank the CNPC and SINOPEC for providing seismic data and expert interpretation results. We also thank the USTC supercomputing center for providing computational resources for this project.

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
