# Peer review of "cigFacies: a massive-scale benchmark dataset of seismic facies and its application"

_Earth System Science Data, 2024_

## Author Comment (AC1)

**Responses to comments from reviewers**

To reviewer 1:

Dear Lorenzo Lipparini,

Thank you for taking the time to provide such detailed and constructive feedback on our manuscript. We greatly appreciate all your thoughtful comments and suggestions, which are helpful to improve the quality and clarity of our work. In response, we have made several modifications to the Introduction, Discussion and Conclusion sections. Additionally, we have carefully rechecked our manuscript to eliminate the repetition and inconsistency of some contents. We have also included more detailed descriptions and evaluations of the classified results, along with a thorough comparison to the expert interpretation results. The related modifications are shown in the responses and marked in the manuscript revision history.

Thanks!

However, in order to improve the manuscript and get to publish it, I would suggest:
1.  - avoid repetitions (some concepts are reported 2 to 4 times)
    Thank you for emphasizing this point. We have carefully rechecked the manuscript and addressed the repetitions of some concepts. For example, we have simplified or changed the expressions to reduce the times of same repetitions (such as, three strategies of …, massive-scale, feature-rich, benchmark dataset of seismic facies)

2.  - limit the introduction to introduce the work only, not to describe it all in summary. Lines from 46 ("Initially, ...) to 50 should be better used to introduce the "Methodology" part (2)
    Thanks for your advice. We have modified this paragraph by removing this section ("Initially, …") and incorporating a description of the current dataset construction methodology: "Currently, the construction of the dataset …". (Lines 44-49)

3.  - expand a bit the discussion and the comparison between predicted results and expert interpretation, as this would be of strong interest for readers, after all the work done to get there. More examples, in section, map and 3D view would be a benefit to the article.
    Thank you for highlighting this important point. We have included additional 2D profile results along the inline direction, as shown in Fig.13 and Fig.14, and provided some additional assessment of the predicted results, including the trend of reef-top interface reflection axis and the accuracy of patch reef. Additionally, we have added the comparison of trends and details between our predicted results and expert interpretation results. (Lines 202-205 and Lines 214-216)

4.  - discuss a bit how the scale of observation and the scale of the observed/classified objects have been considered within the workflow.

We appreciate this feedback. We have expanded our discussion on how to define an appropriate scale for the sliding window in our manuscripts. (Lines 157-190)

5. - more clearly separate discussion from conclusion

Thanks for your suggestion. We have modified the content and structure of the Discussion and Conclusion sections to ensure clearer separation. For example, we have modified the Discussion to offer more insightful and objective analysis of the final dataset construction and predicted results, as well as also providing a deeper examination of the limitations associated with these processes. (Lines 223-247) Besides, we have also revised the Conclusion to not only include a summary of the entire work and its significance but also to incorporate directions for the future research and developments. (Lines 266-271)

6. - the sentences at line 222 (Although the predicted results are roughly consistent with the human interpretation results), and line 249 (..achieves notable performance in seismic facies classification across two distinct 3-D field datasets), appear quite different from the one in line 12 (The predictions are highly consistent with expert interpretation results), and line 221 (Our final sedimentary facies result ... is highly consistent with the expert interpretation of sedimentary facies shown in Fig. 12e). Please consider these inconsistencies

Thank you for offering this thoughtful comment. We have corrected these inconsistencies.

7. - the conclusion could be improved, as they appear as a summary of the work done, not really a conclusion, while part of the conclusion and future developments are reported in the discussions.

Thanks for your suggestion. We have modified the structure and content of the Conclusion, adjusting the contents of future developments from the Discussion section to the Conclusions. (Lines 266-271)

---

## Author Comment (AC2)

**Responses to comments from reviewers**

To reviewer 2:

Dear Tao Zhao,

We sincerely appreciate the detailed and constructive feedback you provided on our manuscript, which have been invaluable in improving the quality and clarity of our work. In response, we have provided additional discussions and comparisons between skeletonized data and seismic amplitude data. Additionally, we have revised and expanded the detailed descriptions of field data application processes, making it more readable. To better assess our results, we have also expanded additional 3D maps of predicted section results from different perspectives. The related modifications are shown in the responses and marked in the manuscript revision history.

Thanks!

1. The choice of using the skeletonized binary mask instead of seismic amplitude as the ML input is very interesting. It certainly highlights the structural information. However, on the other hand, it also lacks the rich information amplitude data carry. I suggest providing more detailed discussion on this choice, ideally with comparison between using seismic amplitude (can be normalized) and skeletonized data.

    Thank you for highlighting this important point. In this work, we ultimately chose to focus on the skeletonized image rather than seismic amplitude data, was mainly based on several considerations:

    (1) The task of this work is to classify the data into major categories such as parallel and subparallel, prograding clinoform, fill, hummocky and chaotic), which is primarily based on the differences in structural information in the geological concept shown in knowledge graph in Figure 2. Therefore, using skeletonization data, which highlights structure features, is sufficient for this task.

    (2) Seismic amplitude, continuity, frequency, and wave patterns actually carry rich information, but they are intended for more refined seismic facies classification (sub-classes within each pattern) and are mainly used for micro-sedimentary analysis or reservoir prediction. Additionally, seismic amplitude will introduce spatial variations within stratigraphic layers, which may reduce the generalization ability of the network across different surveys.

    (3) Skeletonization data provides a standardized mode, ensuring higher consistency in the datasets constructed from three strategies, thus enhancing the generalization and stability of AI model across different surveys. Additionally, using skeletonization data can address the data variations (such as size, amplitude, and frequency), since skeletonization data is binary and broad-band (or full-band) in frequency spectrum, and the resizing process can standardize the size of skeletonization data without affecting the data

appearance and frequency spectrum.

In summary, we chose to focus on skeletonized image to emphasize the geological structural information in seismic data, while minimizing the uncertainties and variations introduced by other seismic attributes. This aligns with our proposed knowledge graph for seismic facies classification, enabling the model to classify seismic facies based on standardized mode, without the challenges posed by seismic amplitude variations.

Undeniably, the rich information carried by seismic amplitude or other attributes are also significant. We will refine this aspect in subsequent work, such as developing a more comprehensive and refined knowledge graph of seismic facies that combines muti-attributes such as geological structure, frequency, amplitude, and continuity features. Guided by this updated graph, we will further construct a more representative benchmark dataset of seismic facies. We have expanded this point in our discussion and conclusion. (Lines 233-236 and Lines 266-269)

2. What is the size of the 2D image patches?

Thanks for your timely reminder. In this work, the size of the 2D image patches is 128(crossline)×128(time). We have expanded a description of this scale in our manuscript. (Line 150, Line 157, and Fig.10)

3. In the synthetic data generation section, it is not clear to me how the examples are categorized if the generation process is a truly random selection among the possible "curve functions", and how the discontinuous events are generated. More details or references on this process will be very helpful.

Thank you for these valuable suggestions. Our descriptions here may be slightly misleading. In second strategy, we first define different combinations of geological structural curves based on different seismic facies categories. Then we set random shape parameters of these curves combine them at random intervals. We have modified these contents in our manuscripts. (Lines 116-118)

The discontinuous events primarily manifest in two ways, one is the discontinuity of geological structural curves, which are realized by randomly applying local masks to each curve. The second one is the unconformities caused by stratigraphic pinch-out are modeled by setting boundaries during the combination of different slope of curves.

4. In the application examples, facies classification is posed as an image classification task, which requires running inference on 2D image patches centered at every voxel in a 3D volume. This can be very computationally expensive. Any reason why pose it as classification but not segmentation, which is far more efficient for this use case?

Thanks for your insightful comments. Honestly, it indeed to be a more efficient approach to treat the seismic facies classification as a segmentation task. However, there are some challenges for segmentation task. For example:

(1) In the case of a 3D segmentation task, the training dataset also needs to be 3D. Collecting and curating a massive-scale 3D field segmentation dataset is

considerably more challenging than working with a 2D field classification dataset.

(2) Due to the supervised learning approach, the manual interpretated labels of a large amount of field datasets for a segmentation task is more tedious and time-consuming than for a classification task.

(3) Constructing synthetic data with corresponding segmentation labels is more complex than with classification labels. Besides, it is particularly challenging for AI-based generation method, as there is no robust approach currently available for generating synthetic data with corresponding segmentation labels.

(4) The generalization of the trained segmentation model across different field surveys is probably weaker than the classification model.

The field data examples miss a lot of details.

5. Two horizons are used to extract the 2D image patches. How does the number samples per trace become the same across all traces in an image patch?

6. If a constant number of samples are used per trace above the base horizon, why do we need the top horizon?

Thank you for raising these important points. The number of samples in the sliding window per trace mainly depends on the average height of the top and bottom horizons across the entire target stratigraphic section. Besides, the sliding window is centered at the midpoint of top and bottom horizons each trace, ensuring the stratum to be classified is located near the center of the extracted image patch. To better clarify these processes, we have modified and expanded the details on how to extract the image patch and how to define an appropriate scale of sliding window in the manuscript. (Lines 187-189)

Besides, the average height setting is a simplification, and a more suitable approach would be to extract the image patch fully bounded by the actual height of each trace. Although this process would result in varying sizes, the introducing of skeletonization data and standardization and skeletonization processes will ensure that all raw images are resized to a uniform size without effecting its appearance and frequency spectrum.

7. In Figure 12 b and f, the windows are tilted following the dip of the formation. Are these the actual windows? If yes, how are data extracted from such dip-oriented windows?

Thanks for your comments. In practice, we first flatten the target stratigraphic section and then extract image patch using the sliding window. Therefore, the windows in Figs.12 b and f actually represent the original positions of the window after flattening. To avoid such confusion, we have modified the order of the flattening and sliding window processes in our manuscript. (Lines 186-190)

8. In Figure 12 c-e, and g-i, it seems like these maps are extracted along the base horizons. Does this mean that the prediction happened on image patches centered at the base horizons? This is different from what is described in the text (extracted

between top and base).

Thanks for your advice. The window is actually centered on the midpoint of the top and bottom horizons. We have expanded these contents in our manuscript. (Lines 187-189) To better display the predicted seismic facies results and corresponding 3D target seismic data, we overlap them with the bottom horizon.

9. From the data description here, it seems like the network performs a segmentation task. But the model description reads like it does classification. Please clarify.

Thanks for your comment. In fact, we use a trained classification network to classify the field data pixel by pixel (or trace by trace) through a sliding window, and ultimately combine the classification results of each pixel (or trace) to produce a segmented seismic facies classification result. (Lines 187-192)

10. Another useful information to share is the amount of overlap for the sliding window.

Thanks for your insight suggestion. Applying the sliding window is pixel by pixel (or trace by trace), so the amount of overlap is primarily determined by the center of each trace in target stratigraphic section.

11. L88. "significant inference" to "significant interference".

Thanks. Corrected. (Line 86)

Figures

12. Figure 4. Please explain 4d in the caption.

Thanks for your advice. We have explained the Fig.4d in the caption. (Fig.4)

13. Figure 12 -14. Need annotation for axes (preferably with range).

Thanks for your suggestion. We have added annotation for axes and textual descriptions of data range for field seismic data and predicted results and sedimentary results. (Lines 196-197, Lines 208-209 and Figs.12-14)

14. If the results can be visualized in vertical sections, I recommend to include additional figures showing the classification results along inline/xline.

Thanks for your constructive suggestion. We have included additional figures along inline direction shown in Figs.13, 14 in the manuscript. (Figs.13g-i and Figs.14g-i)